

# Characterization of Submicron Organic Particles in Beijing During Summertime: Comparison Between SP-AMS and HR-AMS

Junfeng Wang[1,2,*], Jianhuai Ye[2], Dantong Liu[3], Yangzhou Wu[3], Jian Zhao[4], Weiqi Xu[4], Conghui Xie[4], Fuzhen Shen[1], Jie Zhang[5], Paul E. Ohno[2], Yiming Qin[2], Xiuyong Zhao[6], Scot T. Martin[2], Alex K.Y. Lee[7], Pingqing Fu[8], Daniel J. Jacob[2], Qi Zhang[9], Yele Sun[4], Mindong Chen[1] and Xinlei Ge[1,*]

[1]Jiangsu Key Laboratory of Atmospheric Environment Monitoring and Pollution Control, Collaborative Innovation Center of Atmospheric Environment and Equipment Technology, School of Environmental Science and Engineering, Nanjing University of Information Science and Technology, Nanjing, China
[2]School of Engineering and Applied Sciences, Harvard University, Cambridge, MA, United States
[3]Department of Atmospheric Sciences, School of Earth Sciences, Zhejiang University, Hangzhou, China
[4]State Key Laboratory of Atmospheric Boundary Layer Physics and Atmospheric Chemistry, Institute of Atmospheric Physics, Chinese Academy of Sciences, Beijing, China
[5]Department of Atmospheric Science, Colorado State University, Fort Collins, CO, United States
[6]State Environmental Protection Key Laboratory of Atmospheric Physical Modeling and Pollution Control, State Power Environmental Protection Research Institute, Nanjing, China
[7]Department of Civil and Environmental Engineering, National University of Singapore, Singapore
[8]Institute of Surface-Earth System Science, Tianjin University, Tianjin, China
[9]Department of Environmental Toxicology, University of California Davis, Davis, CA, United States

*Corresponding author: Xinlei Ge (Email: caxinra@163.com); Junfeng Wang (Email: wangjunfeng@g.harvard.edu).





**ABSTRACT**
Black carbon (BC) particles in Beijing summer haze play an important role in
regional radiation balance and related environmental processes. Understanding the
factors that lead to variability in the impacts of BC remains limited. Here, we present
observations by a soot-particle aerosol mass spectrometer of BC-containing submicron
particulate matter (BC-PM$_1$) in the summer of 2017 in Beijing, China. These
observations were compared to concurrently measured total non-refractory submicron
particulate matter (NR-PM$_1$) by a high-resolution aerosol mass spectrometer (HR-
AMS). Distinct properties were observed between NR-PM$_1$ and BC-PM$_1$ related to
organic aerosol (OA) composition with hydrocarbon-like OA in BC-PM$_1$ up to two-
fold higher than that in NR-PM$_1$ in fresh vehicle emissions, suggesting that a part of
HOA in BC-PM$_1$ may be overestimated due to the change of the collection efficiency
of SP-AMS. Cooking-related OA was only identified in NR-PM$_1$, whereas aged
biomass burning OA (A-BBOA) was a unique factor only identified in BC-PM$_1$. The
A-BBOA was linked to those heavily coated BC, which may lead to enhancement of
light absorption ability of BC by a factor of two via the "lensing effect". More-oxidized
oxygenated OA identified in BC-containing particles was found to be slightly different
from that observed by HR-AMS, mainly due to the influence of A-BBOA. Overall,
these findings highlight that BC in urban Beijing is partly of agricultural fire origin and,
a unique biomass burning-related OA associated with BC may be ubiquitous in aged
BC-PM$_1$, and this OA may play a role in affecting air quality and climate that has not
previously been fully considered.





**1. Introduction**
Black carbon (BC) is an important component of atmospheric aerosol that exerts
negative effects on regional radiation balance (Bond et al., 2013) and human health
(Janssen, 2012). It absorbs solar radiation, leading to direct atmospheric heating
(Ramanathan and Carmichael, 2008). Indirectly, BC-containing particles (BCc) can
also serve as cloud condensation nuclei upon mixing with hydrophilic species (e.g.,
sulfate), resulting in changes in cloud properties (Wu et al., 2019). Inhalation of BC is
associated with adverse health impacts such as respiratory diseases and birth defects
(Janssen, 2012).
BC particles are released to the atmosphere directly and usually mixed with non-
BC materials (e.g., inorganic and organic) from incomplete fuel combustion and open
fires (Ramanathan and Carmichael, 2008;Bond et al., 2013;Chen et al., 2013). Non-BC
species also can coat onto primary BCc in the atmosphere through condensation and/or
coagulation processes (Lee et al., 2017). These atmospheric processes gradually alter
the mixing state and the morphology (e.g., from an externally-mixed fractal structure
(Buseck et al., 2014) into an internally-mixed "core-shell" structure (China et al., 2015))
of BCc. These alterations can enhance the light absorption capacity of the BC core via
the "lensing effect" due to the increased light absorption cross-section as a result of the
enhanced coating thickness (Saleh et al., 2015;Cappa et al., 2012). Additionally, the
chemical constituents of BCc may dynamically change during the aging processes, also
lead to changes in the light absorption capacity of the particles (Wang et al., 2019;Wang
et al., 2017). Because these physical and chemical processes of both organic and
inorganic species inside BCc continuously alter particle properties throughout the
lifetime of the particles, great uncertainty remains in quantifying the light absorption
ability of BC (Liu et al., 2018;Liu et al., 2019). Understanding the relationship of
mixing state and chemical composition to the light absorption properties of BCc, as
well as its spatiotemporal distribution, is of importance to accurately evaluate the
impacts of BC in regional air quality.
Aerodyne high-resolution aerosol mass spectrometry (HR-AMS) (Canagaratna et
al., 2007) has been widely applied in field studies to investigate the chemically-resolved
composition of non-refractory submicron particulate matter (NR-PM$_1$, species that
vaporize at temperature < 600 °C)(Li et al., 2015;Lee et al., 2013;Sun et al., 2012;Ge
et al., 2012b;Ge et al., 2012a;Xu et al., 2019c;Sun et al., 2014). However, the working
temperature of the standard HR-AMS tungsten vaporizer (600 ºC) is not sufficient to



vaporize refractory species such as BC. To overcome this limitation, soot-particle
aerosol mass spectrometry (SP-AMS) is developed (Onasch et al., 2012). In addition to
the standard tungsten vaporizer, SP-AMS is equipped with a laser vaporizer (with a
wavelength of 1064 nm) which selectively heats BC (core), together with the non-BC
species mixed with it (Wang et al., 2016). This novel technique makes it possible to
compare the compositions of submicron BCc (BC-PM$_1$) and NR-PM$_1$, allowing a more
accurate assessment of the impacts of BC. However, a question is whether the ion
fragments of organic species ionized by the 70eV electron impact of SP-AMS and HR-
AMS are the same in terms of different thermal schemes. It has been reported that the
mass spectra of NR-PM$_1$ organic have high m/z 44 (mainly $CO_2^+$) signal, while the
mass spectra of BC-related organics have high m/z 43 ($C_3H_7^+$ and $C_2H_3O^+$) signal. The
reason for this is the SP-AMS provides vaporization of the BC-PM$_1$ at lower
temperatures compared to the standard tungsten vaporizer of the HR-AMS, resulting in
less overall fragmentation and therefore less $CO_2^+$ production in the laser, in addition,
the lower fragmentation also causes the presence of more ion fragments at m/z > 100
amu in the SP-AMS mass spectra compared to that of HR-AMS (Canagaratna et al.,
2015b;Massoli et al., 2015). Nevertheless, quantification of BC-PM$_1$ organic aerosol
(OA) factors identified from positive matrix factorization (PMF) has been reported that
were not significantly affected by the differences of mass spectra between HR-AMS
and SP-AMS (Lee et al., 2017;Massoli et al., 2015).
To date, there have only been a few studies that have compared the differences of
species in BC-PM$_1$ and NR-PM$_1$ (Lee et al., 2017;Collier et al., 2015;Massoli et al.,
2015). Lee et al. found that cooking-related organic aerosol (COA) may externally mix
with BC in summertime California (Lee et al., 2017). The COA factor was identified in
NR-PM$_1$ organic aerosol (OA), but not in the BC-related OA. Wang et al found that
transported biomass burning organic aerosol could be thickly coated on BC in central
Tibetan Plateau and significantly enhance the light absorption capacity of BC cores
(Wang et al., 2017). Interestingly, the transported biomass burning organic aerosol was
not resolved in NR-PM$_1$ OA particles from concurrent HR-AMS measurements (Xu et
al., 2018). These studies suggest that BC-related OA may undergo different
atmospheric processes compared to those do not contain BC.
Beijing is a megacity known for high particulate matter (PM) concentrations. BC-
PM$_1$ during haze events of summertime Beijing may have distinct sources and
properties than other locations in the world. As a part of the UK-China Air Pollution





and Human Health (APHH) project summer campaign (Shi et al., 2019), in this study,
we focus on the differences of individual species between BC-PM$_1$ and NR-PM$_1$
regarding their chemical composition, mass loadings, sources, and formation pathways
in summertime in urban Beijing. Results from this study provide a better understanding
of the formation mechanism of OA particles in Beijing haze and valuable insights in
assessing their impacts on air quality.

**2. Experiments**
**2.1. Sampling site and period**

The observations were conducted at a rooftop laboratory (8 m above ground level)

in the Tower Division of the Institute of Atmospheric Physics (IAP), Chinese Academy
of Sciences (CAS) in urban Beijing (39º58′N, 116º22′E), China, from 4 to 29 June,
2017. This site has been reported multiple times to be a typical urban observation
location (Xie et al., 2019b;Liu et al., 2019;Wang et al., 2019;Qiu et al., 2019;Xu et al.,
2019a;Xie et al., 2019a). The site is located around the North 3$^{rd}$ Ring Road of Beijing.
A highway is approximately 360 m to the east and a lot of restaurants (e.g., Sichuan
style and BBQ) are within 100 m on the north side.

**2.2. Instrumentation**

Two Aerodyne Aerosol Mass Spectrometers (AMS), including a laser-only Soot-

Particle AMS (SP-AMS) and a High-Resolution Time of Flight AMS (HR-AMS) were
deployed to measure chemical compositions and size distributions of BC-PM$_1$ and NR-
PM$_1$, respectively. Three types of species were measured during the campaign: NR-
PM$_1$, including BC-free species (Type I) and non-refractory species that mixed with BC
(Type II), and BC-PM$_1$ (BC core and species coated on the core)(Type III). HR-AMS
is capable of measuring Type I and Type II, while laser-only SP-AMS can measure
Type II and Type III. A shared PM$_{2.5}$ cyclone inlet (Model URG-2000-30ED) with 3
Lpm flowrate and a diffusion dryer were used prior to the sampling. The detailed
information on the operation of HR-AMS and SP-AMS during the sampling campaign
can be found in previous literature (Xie et al., 2019a;Xu et al., 2019d). Details of tuning,
calibration, and configurations of the two AMS instruments can be seen in our previous
papers (Wang et al., 2019;Xu et al., 2019a;Xu et al., 2019d). Mixing ratios of O$_3$, and
NO$_2$ (Thermo Fisher Scientific, model 49$i$ and model 42C) were measured in parallel
simultaneously. Vertical meteorological parameters, including temperature ($T$) and





relative humidity (*RH*), were measured from the IAP 325m meteorological tower.

**2.3. Data Analysis**
AMS data analysis was performed by using Squirrel 1.57 and Pika 1.16I based on
Igor Pro 6.37 (WaveMetrics Corp.). The measurement of filtered air was performed for
24 hours before the start of the campaign to determine the detection limits of various
aerosol species and to adjust the fragmentation table. The relative ionization efficiency
(RIE) of BC was calibrated with Regal Black (RB, REGAL 400R pigment black, Cabot
Corp.). The average ratio of $C_1^+$ to $C_3^+$ ionized from pure BC (RB) was determined to
be 0.53, which minimizes the influence of $C_1^+$ from non-refractory organics. The RIE
of BC was determined to be 0.17 based on calibrations performed before, in the middle,
and at the end of the campaign. RIEs of $NO_3^-$, $SO_4^{2-}$, $NH_4^+$ were determined to be 1.1,
0.82, and 3.82, respectively, and default values of 1.3 and 1.4 for RIEs of Chl and Org
were applied, respectively (Canagaratna et al., 2007). Consistent with BC-$PM_1$
measurements in previous studies, the RIEs calibration of $NO_3^-$, $SO_4^{2-}$, $NH_4^+$ were
performed before the tungsten vaporizer was removed, by assuming those RIEs remain
unchanged throughout the campaign (Wang et al., 2017). Polystyrene latex (PSL)
spheres (100-700 nm) (Duke Scientific Corp., Palo Alto, CA) were used to calibrate the
particle size distribution before the campaign. The collection efficiency (CE) of 0.5
were applied for both HR-AMS and SP-AMS in this study. It should be noted that, the
BC quantification will not be affected by particle bouncing without the tungsten
vaporizer, which could affect the CE in the standard HR-AMS measurements
(Canagaratna et al., 2007). However, the CE will be governed by the overlap of particle
beam and laser beam (Lee et al., 2017;Massoli et al., 2015;Willis et al., 2014). Both
HR-AMS and SP-AMS resolved mass concentrations of NR-$PM_1$ and BC were
calculated based on V-mode high-resolution fitting. Due to different vaporization
schemes between the HR-AMS and SP-AMS, mass spectra from these two instruments
even for the same population of aerosols are not entirely the same. Because laser-only
SP-AMS generally results in less overall fragmentation, its mass profile may contain
more large *m/z* fragments and less small *m/z* fragments compared to that from HR-
AMS(Massoli et al., 2015). In addition, the elemental ratios of organics reported here,
i.e., oxygen-to-carbon and hydrogen-to-carbon ratios (O/C and H/C) were calculated
based the "Improved-Ambient (I-A)" method(Canagaratna et al., 2015a).
Positive matrix factorization (PMF)(Paatero and Tapper, 1994) was performed on





the high-resolution organic mass spectra matrix of both NR-PM$_1$ and BC-PM$_1$ (e.g., BC
(C$_x^+$), and species associated with BC) across m/z 12–120 using PMF Evaluation Tool
written in Igor (Ulbrich et al., 2009), following the standard procedure (Zhang et al.,
2011). Four types of organic aerosol (OA) from total NR-PM$_1$ (see our previous
paper)(Xu et al., 2019c) and five OA factors from BC-PM$_1$ were identified. C$_x^+$ was
involved in the calculation of elemental ratios (e.g, O/C and H/C) of PMF OA factors.
All data presented in this paper were averaged hourly and are presented at local time
(Beijing Time, UTC+8).

**3. Results and discussion**
**3.1. Overview of observations**

Figure 1 shows the temporal variations of selected chemical species during the

campaign. Information for other variables is provided in the supplementary materials
(SM). The two cases labeled in Figure 1 are of interest. Case I (June 8-13) was
characterized with high NO$_2$ concentrations (average $26.7 \pm 13.5$ ppb, Table S1) and
relatively low O$_3$ concentrations ($41.7 \pm 30.0$ ppb) with NO$_2$-to-O$_3$ ratio of 0.64. Case
II (June 17-22) was featured by low NO$_2$ ($14.9 \pm 5.9$ ppb) and high O$_3$ ($84.6 \pm 30.6$ ppb)
concentrations with an NO$_2$-to-O$_3$ ratio of 0.18. Unlike winter Beijing haze pollution,
*RH* remained at a relatively low level ($36.5 \pm 15.3\%$), which is not expected to play a
significant role in OA formation during the campaign (Figure 1b and Figure S1). In
contrast, a strong correlation has been observed between temperature and O$_3$ ($r^2 = 0.53$).
The temperature was higher on average in Case II ($29.8 \pm 3.8$ ºC) than in Case I (26.1
$\pm 4.1$ ºC).

The mass concentrations and mass concentration ratios of organic (Org), sulfate

(SO$_4^{2-}$) and nitrate (NO$_3^-$) in NR-PM$_1$ (in solid line) and BC-PM$_1$ (in dotted line) are
shown in Figures 1c-e. High correlations were observed between BC-PM$_1$ and NR-PM$_1$
measurements for SO$_4^{2-}$ ($r^2 = 0.70$) and NO$_3^-$ ($r^2 = 0.86$), but not for Org ($r^2 = 0.49$).
This result suggests that, BC-PM$_1$ Org has distinct sources or formation pathways from
NR-PM$_1$ Org. Comparing two cases, the average mass ratios of BC-PM$_1$ to NR-PM$_1$ for
SO$_4^{2-}$ and NO$_3^-$ in Case I ($0.24 \pm 0.11$ and $0.37 \pm 0.12$) were close to those in Case II
($0.19 \pm 0.06$ and $0.31 \pm 0.07$). However, ratios of BC-PM$_1$ to NR-PM$_1$ for Org were a
factor of greater for Case I ($0.74 \pm 0.32$) compare to Case II ($0.46 \pm 0.13$). During the
nighttime, this ratio increases to almost unity in Case I. Additionally, BC concentration
in Case I (average $2.6 \pm 1.6$ µg m$^{-3}$) was 1.5 folds higher than in Case II (average $1.7 \pm$





$0.8\mu g\ m^{-3}$). The implication is that the organic is mostly associated with BC and likely
comprised of freshly emitted compounds in Case I. This is also evident by the moderate
correlation between $NO_2$ and BC-$PM_1$ Org ($r^2 = 0.42$) in Case I. On the other hand, the
lower Org ratio in Case II with higher $O_3$ concentrations indicates greater oxidation and
secondary processes in non-BC particles.

**3.2. Source apportionment of BC-$PM_1$ OA**

To further investigate the differences between organics in NR-$PM_1$ and BC-$PM_1$,

the comparison of PMF OA factors between NR-$PM_1$ and BC-$PM_1$ Org is necessary.
Four factors were identified from PMF analysis of the NR-$PM_1$ Org matrix, including
hydrocarbon-related OA (HOA), cooking OA (COA), less-oxidized oxygenated OA
(LO-OOA), and more-oxidized oxygenated OA (MO-OOA). Details of the NR-$PM_1$
PMF analysis can be found in our previous study (Xu et al., 2019d). Here we only
present the PMF results of the SP-AMS measured BC-$PM_1$ Org. As shown in Figure 2,
five factors were resolved by PMF with factors including a HOA, a less oxidized OOA
(OOA1), three more-oxidized OOA factors were recombined into one OOA factor
(MO-OOA= Aged- biomass burning organic aerosol (A-BBOA) + OOA2 + OOA3).
Diagnostic plots of this PMF solution is presented in Figure S2.

HOA consists of a series of hydrocarbon fragments ($C_xH_y^+$) in its mass spectrum

(Figure 2f), thus having a low O/C ratio (0.13) but high H/C ratio (1.62). It has a $r^2$ of
0.92 with $C_4H_9^+$ (m/z = 57) and a $r^2$ of 0.57 with $NO_x$ (Figure 2a), indicative of its
sources from vehicle emissions(Xu et al., 2019b). It also correlated tightly with BC ($r^2$
of 0.70) and a series of polycyclic aromatic hydrocarbons (PAHs) ions, e.g., $C_9H_7^+$ (m/z
115, $r^2$ of 0.63).

The second factor has a remarkably high fraction of the biomass burning organic

aerosol (BBOA) marker ions of $C_2H_4O_2^+$ (m/z = 60) (1.31%) and $C_3H_5O_2^+$ (m/z = 73)
(1.34%) in its mass spectrum (Figure 2g), much higher than that observed in non-BBOA
(e.g., 0.3% at m/z = 60) in previous studies (Sun et al., 2016;Xu et al., 2019b;Wang et
al., 2017). As expected, the temporal variation of this factor correlated tightly with those
of $C_2H_4O_2^+$ and $C_3H_5O_2^+$ ($r^2$ of 0.71 and 0.72, respectively). In addition, the mass
spectrum of this factor is strikingly similar to that of the transported BBOA which was
observed at a remote site in the central Tibetan Plateau (Wang et al., 2017), with a $r^2$ of
0.97. Here we categorized the transported BBOA as aged-BBOA (A-BBOA) identified
in this study. Similar to the A-BBOA observed in Tibetan Plateau, which has an O/C



ratio of 0.51, this factor also has a relatively high O/C ratio of 0.48, greater than that of
primary BBOA (O/C of 0.18−0.26)(Wang et al., 2017). These findings support that the
second factor may be associated with the oxidation of biomass burning emissions. The
temporal variation of ABBOA in the Tibetan Plateau was reported to be highly
correlated with the potassium ion fraction ($K^+$, $r^2$ of 0.78), and $K_3SO_4^+$ ($r^2$ of 0.92).
However, the temporal variation of the second factor in this study is only correlated
well with that of $K_3SO_4^+$ ($r^2$ of 0.64) but not $K^+$ ($r^2$ of 0.01). The reason for this
phenomenon is that the major source of $K^+$ in remote sites like the Tibetan Plateau was
long-distance transport of $K_2SO_4$ particles, which probably from biomass burning-
related K-containing salts interacts with $H_2SO_4$ (V. Buxton et al., 1999). In contrast,
there are multiple primary sources of $K^+$ in $PM_1$ (e.g., diesel-vehicle emissions, and
mainly KCl particles) in urban areas (Figure S3). Based on these observations, $K_3SO_4^+$
could be defined as an external A-BBOA indicator. Moreover, a previous transmission
electron microscopy study also shown that significant agricultural BBOA was mixed
with soot and transport from the North China Plain to urban Beijing, meanwhile, $K_2SO_4$
was also identified within those impact single BBOA-soot particles (Li et al., 2010).
Hence, this second factor is identified as A-BBOA that was subjected to oxidation
during transport to the measurement area as presented in the fire-point map and three-
day back trajectories (Figure S4). June should be the month of maximum agricultural-
related biomass burning in the North China Plain, although we thought that this burning
had been banned in recent years because of air quality concerns (Shen et al., 2019). The
implication is that the effectiveness of banning straw burning may be overestimated.

The OOA1 factor has an O/C of 0.28 (Figure 2h). Similar to the NR-$PM_1$ LO-

OOA(Xu et al., 2019c), it is highly correlated with $C_2H_3O^+$ ($r^2$ of 0.72). The $C_2H_3O^+$
ion (m/z = 43) is an important component of secondary organic aerosol (SOA)(Collier
et al., 2015;Ng et al., 2011) and the diurnal patterns of the OOA1 and $C_2H_3O^+$ both
show a great enhancement around noontime (Figure S5), indicating the importance of
secondary formation of less oxidized organic aerosol through daytime photochemical
activity.

The OOA2 factor has an O/C of 0.42 (Figure 2i) and the OOA3 factor has a smaller

O/C of 0.32 (Figure 2j). OOA2 correlated strongly with sulfate ($r^2$ of 0.92; Figure 2d)
and OOA3 correlated highly with nitrate ($r^2$ of 0.97; Figure 2e). These features agree
well with the previously observation for low-volatility OOA (sulfate-related OOA) and
semi-volatile OOA (nitrate-related OOA) in Tibetan Plateau (Wang et al., 2017).





### 3.3. Comparison of NR-PM$_1$ and BC-PM$_1$ OA factors


The sum of the above-mentioned BC-PM$_1$ A-BBOA, OOA2, and OOA3 fractions
is comparable to the NR-PM$_1$ MO-OOA factor, based on their high O/C ratios. Figures
3a-c are comparisons of the mass loadings of HOA, LO-OOA, and MO-OOA in both
NR-PM$_1$ and BC-PM$_1$. NR-PM$_1$ HOA, LO-OOA, and MO-OOA are strongly correlated
with their counterpart fractions of BC-PM$_1$, with $r^2$ values of 0.68, 0.60, and 0.61,
respectively. In Case I, most of the time, the mass loadings of BC-PM$_1$ HOA and MO-
OOA are higher than those in NR-PM$_1$, while LO-OOA shows the opposite trend. In
Case II, the mass loadings of BC-PM$_1$ HOA are also generally higher than those of NR-
PM$_1$ HOA, however, NR-PM$_1$ MO-OOA and LO-OOA are almost two folds higher than
those of BC-PM$_1$. Figures 3d-f are comparisons of the fractions of HOA, LO-OOA, and
MO-OOA in NR-PM$_1$ and non-BC material in BC-PM$_1$ (coatings), respectively. In Case
I, the fractions of HOA and MO-OOA internally-mixed with BC are almost two times
and four times higher, respectively, than those in NR-PM$_1$, whereas the two LO-OOA
fractions closely track each other. In Case II, two LO-OOA fractions are still overlapped,
but compared to Case I, the fraction of HOA in BC-PM$_1$ coatings is over four times that
of NR-PM$_1$ HOA, and the difference between the two MO-OOA fractions is smaller.
As shown in Figure 4, the average of BC-PM$_1$ HOA fractions ($0.27 \pm 0.17$ and $0.11$
$\pm 0.07$, respectively) are higher than those in NR-PM$_1$ ($0.12 \pm 0.08$ and $0.02 \pm 0.02$,
respectively) in both Case I and Case II, indicating that HOA particles is more internally
mixed with BC compared to other OA materials. However, the possibility that RIE of
OA coating may be lower than the default RIE value should also be considered.
The average mass loadings of NR-PM$_1$ LO-OOA in both Case I and Case II were
higher than those of BC-PM$_1$. However, the fraction of LO-OOA in both NR-PM$_1$ and
BC-PM$_1$ coatings were very close to each other during the two cases, with an average
value of $0.23 \pm 0.10$ and $0.25 \pm 0.12$, respectively, indicating that the probability of LO-
OOA condensation onto the two different types of particles is similar.
A greater difference between the MO-OOA fractions in NR-PM$_1$ and BC-PM$_1$ was
observed in Case I than in Case II, and there is more MO-OOA in BC-PM$_1$ than in NR-
PM$_1$ in Case I. A similar comparison between NR-PM$_1$ MO-OOA with BC-PM$_1$ MO-
OOA without A-BBOA can be found in SI (Figure S6), which shows closer fractions in
both Case I and Case II. Therefore, one possibility which may cause higher MO-OOA
fraction in BC-PM$_1$ than that in NR-PM$_1$ in Case I is the presence of the BC-PM$_1$ A-
BBOA, which is only identified from the BC-PM$_1$ OA. More details of the BC-PM$_1$ A-



BBOA are discussed in Section 3.4.

**3.4. Characteristics of A-BBOA in BC-containing PM$_1$**
Figure 5 shows the high-resolution mass spectra of A-BBOA observed in Nam Co
(June 2015) and Beijing (June 2017) by laser-only SP-AMS. A mass spectra very
similar to that observed in Beijing was also observed in Nanjing (February 2017)(Wu
et al., 2019), with a $r^2$ of 0.95. The A-BBOA observed in Nam Co (the Tibetan Plateau)
was found in the thickest coated and internally-mixed BC-PM$_1$ (the mass ratio of
coatings to BC core ($R_{BC}$) can reach 14), which enhances the light absorption ability
($E_{abs}$) of the BC core by a factor of 1.5 to 2.0 via the "lensing effect".
As shown in Figure 6, A-BBOA was associated with those large particles ($D_{va}$ >
300nm) which were also heavily-coated ($R_{BC}$ > 9, Figure 6a and 6c). Because A-BBOA
is a moderately aged OA, the OSc was very steady when $R_{BC}$ > 9 (Figure 6c). Figure
6b presents the fractions of the OA factors (left) and the degree of light absorption
enhancement ($E_{abs}$, estimated by the mass ratios of BC measured by Aethalometer
model 33 and SP-AMS), as a function of $R_{BC}$. Figure 6d shows the temporal variations
of the fractions of NR-PM$_1$ OA and BC-PM$_1$ OA from 15:00 to 24:00 on June 17, 2017
when the highest A-BBOA concentrations were observed. There is a significant
enhancement of A-BBOA which may account for up to 60% of the total OA coatings,
which could enhance the BC-PM$_1$ MO-OOA fraction (within the purple frame in the
bottom panel of Figure 6d).
In this study, A-BBOA was only observed by SP-AMS and was indeed only
associated with BC. It is likely that A-BBOA was emitted together with BC when
burning biomass fuel, and was oxidized subsequently during the transport. However,
we cannot exclude the possibility that A-BBOA can be detected by HR-AMS. For
example, it might be included in NR-PM$_1$ MO-OOA factor. Without separating A-
BBOA from other organic species, the source apportionment for HR-AMS may obscure
air-quality- and climate-related implications of A-BBOA in the atmosphere, such as the
enhancement of aerosol light absorption ability (Figure 6b).



**4. Conclusions and implications**


Online chemical characteristics of BC and its associated species was for the first
time elucidated in urban Beijing in summer, and compared with those of NR-PM$_1$
species. The biggest difference between the two measurements was in the composition
of the organic species. In particular, we found BC in urban Beijing in June is partially
of agricultural fire origin and, an unique biomass burning-related OA factor (A-BBOA)
which was moderately aged, only existed in thickly coated BC-PM$_1$ ($R_{BC} > 9$), but not
NR-PM$_1$. The unique A-BBOA could make up a significant portion of BC coating
material. In addition to Beijing, similar A-BBOA was also identified in other locations,
such as central Tibet Plateau (Wang et al., 2017) and Nanjing (Wu et al., 2019),
suggesting that it may be ubiquitously present in BC-PM$_1$ in ambient atmosphere.
BBOA species are known to constitute a large portion of light-absorbing organics
(brown carbon, BrC). The delay of BBOA oxidation and its longer duration time on BC
cores can extend the impacts of BC. Moreover, together with our previous study of BC-
associated A-BBOA in Tibet, results presented herein demonstrate that A-BBOA could
lead to thick coating on BC cores, meaning a significant "lensing effect" to the
enhancement of BC light absorption (Liu et al., 2017). As a key component of BC
coating, presence of this factor may also alter the bulk hygroscopicity of BC-PM$_1$. It
could therefore affect its ability as cloud condensation nuclei (CCN)(Wu et al., 2019).
Overall, the emission, evolution and transport of such A-BBOA, may influence the
atmospheric behaviors and influence the role of BC in the air quality and climate (e.g.,
radiative forcing and precipitation). We propose that future laboratory, field, and
modeling studies are needed to verify the presence of A-BBOA, and to evaluate the
regional environmental impacts of it.

*Data availability.* The data in this study are available from the authors upon request
(caxinra@163.com).

*Supplement.*

*ACKNOWLEDGMENTS*

The authors from PRC acknowledge support from the National Natural Science
Foundation of China (21777073) and the National Key Research and Development
Program of China (No. 2018YFC0213802). The authors from Harvard and NUIST





acknowledge additional support through the Harvard-NUIST Joint Laboratory for Air
Quality and Climate (JLAQC).

**ABBREVIATIONS**

BC Black carbon

$PM_1$ Particulate matter with an aerodynamic diameter smaller than 1 μm

$NR-PM_1$ non-refractory $PM_1$

$BC-PM_1$ BC-containing particles in $PM_1$

BrC Brown carbon

HR-AMS High-resolution aerosol mass spectrometer (Aerodyne Research Inc.)

SP-AMS Soot-particle aerosol mass spectrometer (Aerodyne Research Inc.)

IE Ionization efficiency

RIE Relative ionization efficiency

HRMS High-resolution mass spectra

PMF Positive matrix factorization

OA Organic aerosol

SOA Secondary organic aerosolO/C Oxygen-to-carbon ratio

H/C Hydrogen-to-carbon ratio

A-BBOA Aged biomass burning organic aerosol

SV-OOA Semi-volatile oxygenated organic aerosol

LV-OOA low-volatility oxygenated organic aerosol

MO-OOA more-oxidized oxygenated organic aerosol

LO-OOA less-oxidized oxygenated organic aerosol

$R_{BC}$ mass ratio of BC coatings to BC

$D_{va}$ Vacuum aerodynamic diameter





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

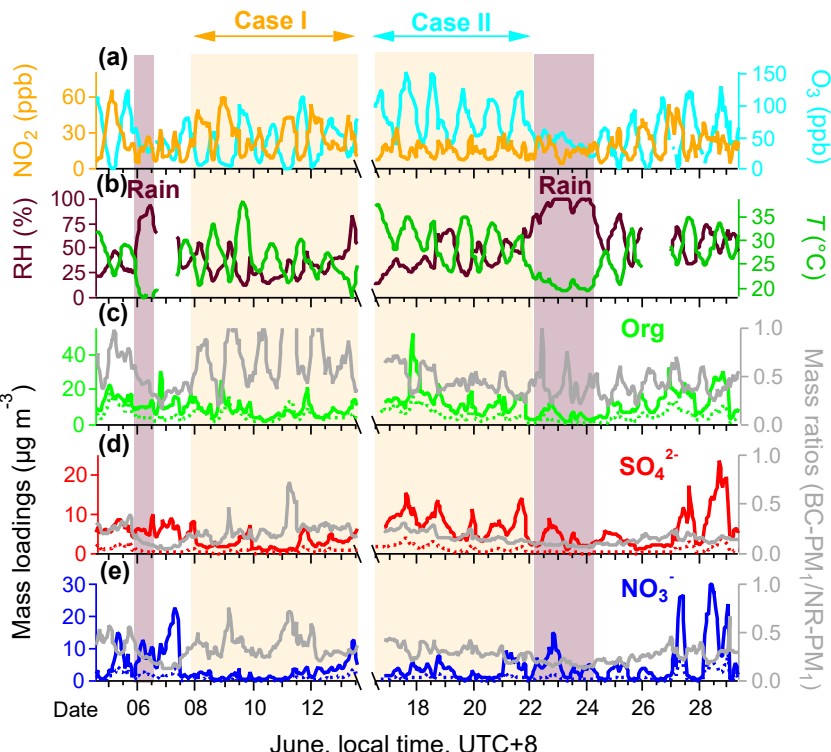


**Figure 1.** Temporal variations of selected chemical species measured in Beijing on June 4 -29, 2017.
(a) mixing ratios of nitrogen dioxide (NO₂) and ozone (O₃); (b) 15-m relative humidity (*RH*) and
temperature (*T*); (c-e) on the left are the mass loadings of organic (Org), sulfate (SO₄²⁻) and nitrate
(NO₃⁻) measured by HR-AMS and SP-AMS, and on the right are mass ratios of individual BC-PM₁
species to NR-PM₁ species (e.g., BC-PM₁ Org to NR-PM₁ Org). The NR-PM₁ species measured by
HR-AMS is in solid line, and the BC-PM₁ species measured by SP-AMS is in the dotted line. The
shaded areas are raining periods. The observation period is divided into two cases according to the
mixing ratio of nitrogen NO₂, Case I and Case II, which represent high NO₂ and low NO₂ mixing
ratios, respectively.

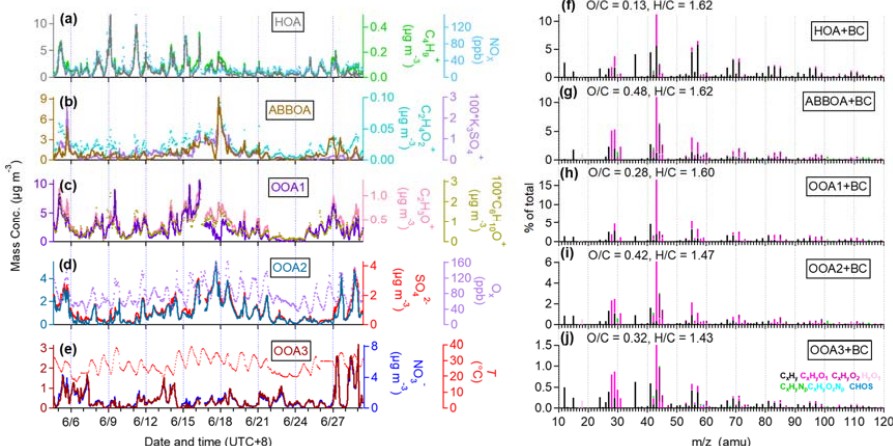

**Figure 2.** Temporal variations (left panels), high-resolution mass spectra (right panels) of five OA factors in summer 2017: (a) and (f) HOA, (b) and (g) A-BBOA, (c) and (h) OOA1 (LO-OOA), (d) and (i) OOA2, and (e) and (j) OOA3. Also shown in the left panels are the time series of other tracers, including $C_4H_9^+$, NOx, $C_2H_4O_2^+$, $K_3SO_4^+$, $C_6H_{10}O^+$, $C_2H_3O^+$, $SO_4^{2-}$ and $NO_3^-$.

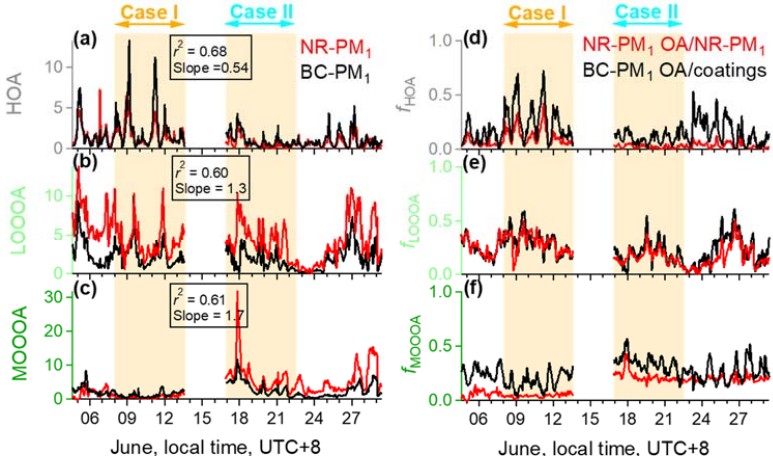

**Figure 3.** Temporal variations of NR-PM$_1$ and BC-PM$_1$ (a-c) HOA, LO-OOA, and MO-OOA (left

panels) and (d-e) their fractions. NR-PM$_1$ OA factors are in red, and the BC-PM$_1$ OA factors are in

black. Here BC-PM$_1$ MO-OOA is the sum of A-BBOA, OOA2 (sulfate-related OOA), and OOA3

(nitrate-related OOA).



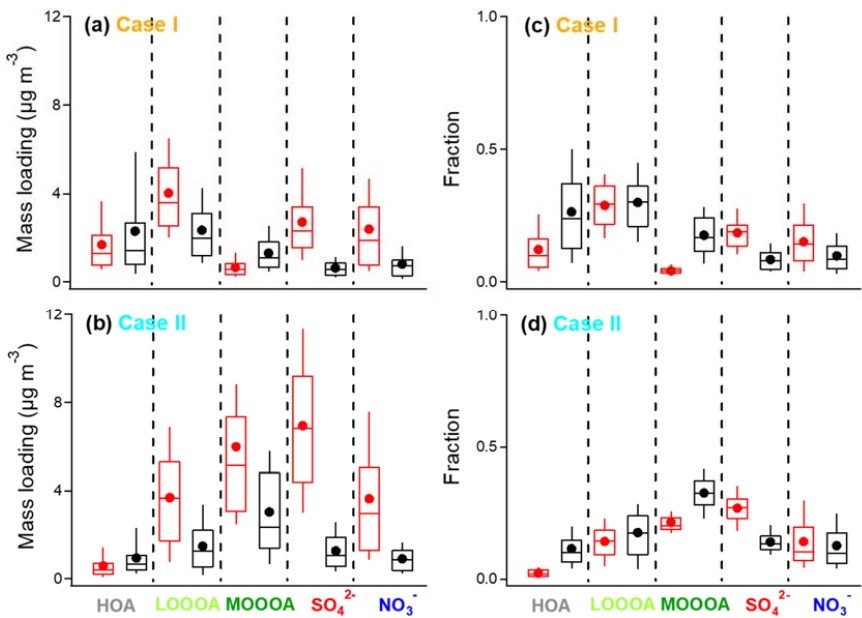

**Figure 4.** Box plots of mass loadings and fractions of five selected species (HOA, LO-OOA, MO-

OOA, $SO_4^{2-}$, and $NO_3^-$) in Case I and Case II. The bounds of boxes represent quartiles, the whiskers

indicate the 90th and 10th percentiles, and the lines and dots inside the boxes are median and mean

values. NR-$PM_1$ OA factors are in red, and the BC-$PM_1$ OA factors are in black.



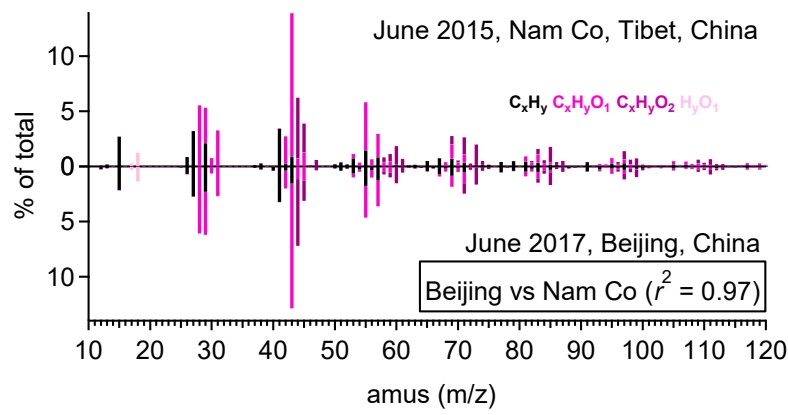

**Figure 5** Comparison between the high-resolution mass spectra of A-BBOA obtained in Nam Co

(June 2015) and Beijing (June 2017).

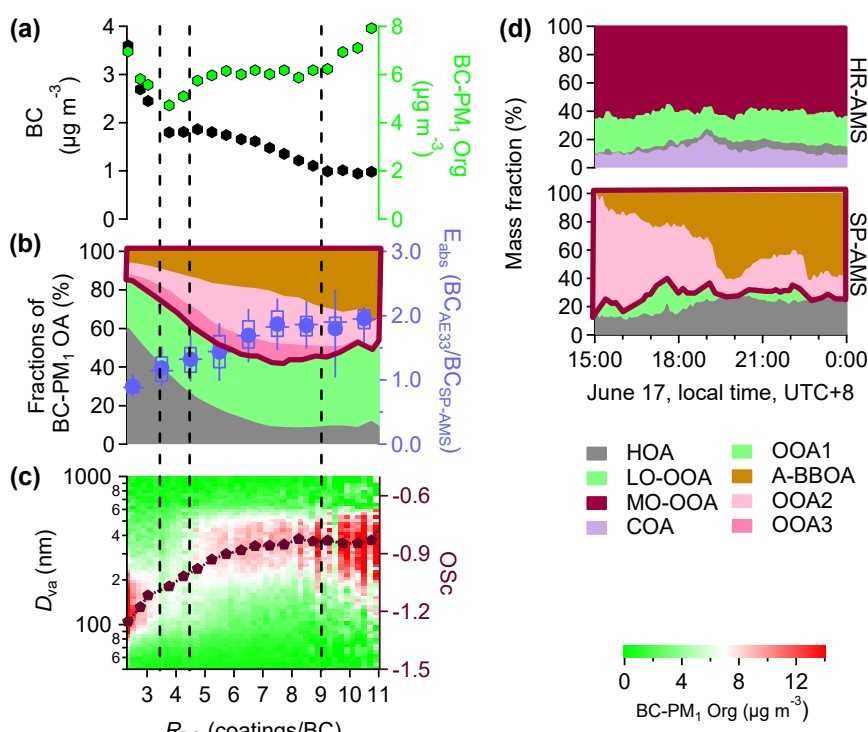

**Figure 6.** (a-c) the mass loadings of BC, BC-PM$_1$ Org, fractions of BC-PM$_1$ OA factors, $E_{abs}$, the oxidation state (OSc = 2*(O/C) – (H/C)) of BC-PM$_1$ Org, and the size distribution of BC-PM$_1$ Org as a function of coating thickness ($R_{BC}$). (d) temporal variations of OA fractions of NR-PM$_1$ and BC-PM$_1$ from 15:00 to 24:00 on June 17, 2017.