# Peer review of "Characterization of Submicron Organic Particles in Beijing During Summertime"

_Atmospheric Chemistry and Physics, 2020_

## Referee Comment (RC1) · Anonymous Referee #1 · 16 Aug 2020

This manuscript presents chemical analysis results of submicron organic aerosols in Beijing during summer. It mainly uses two types of aerosol mass spectrometers and compares the measurement results with each other. Due to different detection schemes, the authors found that the OA determined by SP-AMS are quite different from that of HR-AMS OA. In particular, vehicle-related OA might be detected more by SP-AMS; cooking OA, was not associated with BC; a unique biomass burning OA, on the other hand, was only significantly observed on BC cores. The work provides valuable contribution into understanding the chemical behaviors and therefore the impacts on air quality and climate of OA. It can be accepted for publication in ACP, this reviewer has however a few minor comments as listed below: (1) There are a few typos, gram-

[Figure]

mar or format errors in the manuscript that should be corrected, for example, in Line 38, Line 67, Line 79, Line 235-236, Line299, etc. (2) Line 114: it is not clear what is the "BC-free species" referring to. (3) Line 114-116: Explain a bit more why HR-AMS can measure Type I and II, and SP-AMS for Type II and III. (4) Line 150-155: Did you perform corrections, for example, on elemental ratios, between the two AMS as you state there could be some mass spectral differences due to measurement schemes? (5) Line 248: I suggest to delete this sentence. (6) Line 280-282: As you state that HOA quantification might be influenced by the changes of collection efficiency. Can you explain a bit more about the possible influences of the collection efficiency on other OA factors? (7) Line 320: Is it possible that the A-BBOA fraction (for example, <5%) in total NR-PM1 is too low to be resolved by the PMF? (8) Figure 6: It is better to put the legends adjacent to the HR-AMS and SP-AMS plots directly in d, to make it clear.

---

## Referee Comment (RC2) · Anonymous Referee #2 · 17 Aug 2020

Wang et al. compare the OA properties by parallel measurements using SP-AMS and HR-AMS respectively, in summer Beijing. The AMS technique is suitable for online quantification of OA and in particular SP-AMS can provide a unique piece of OA that coated on rBC cores. The findings are therefore unique and valuable to understand the OA composition and chemistry in megacities like Beijing. The overall interpretation of the data is reasonable and the paper is well written, I suggest its acceptance in ACP after the following minor issues are well addressed.

(1) During the APHH campaign, another type of mass spectrometer (single particle mass spectrometry) was used to elucidate the OA properties too. Some studies should

be included here to facilitate the interpretation. (2) Some typos or citation formats do not follow the ACP style, please check and revise. (3) In the instrumentation section, some necessary technical details are missing. For example, what is the m/z range of the OA mass spectra for HR-AMS and SP-AMS? Time resolution? Operation modes (V or W?) Is the tungsten vaporizer physically removed or turned off in SP-AMS? (4) Xie et al (Atmos Environ 2019;213:499-504) shows different PMF results from this study, is it because the datasets used for PMF analysis are different? (5) Line 316-319: The ABBOA is not separated in HR-AMS dataset, is it likely because that the ABBOA contains more refractory components? (6) References: Line 504-516, the references are the same, but repeated twice. (7) Figure 5: There are only four ion families here. How about the nitrogen-containing organic ions, although they may have little influences?

---

## Author Comment (AC1) · 6 Oct 2020

**Responses to reviewer comments**

We thank the reviewers for their detailed, helpful, and overall supportive comments. We have revised the manuscript to account for each comment. Responses to the individual comments are provided below. Reviewer comments are in **bold**. Author responses are in plain text. Modifications to the manuscript are in *italics*. Line numbers in the response correspond to those in the revised manuscript text file.

**Reviewer #1**
**This manuscript presents chemical analysis results of submicron organic aerosols in Beijing during summer. It mainly uses two types of aerosol mass spectrometers and compares the measurement results with each other. Due to different detection schemes, the authors found that the OA determined by SP-AMS are quite different from that of HR-AMS OA. In particular, vehicle-related OA might be detected more by SP-AMS; cooking OA, was not associated with BC; a unique biomass burning OA, on the other hand, was only significantly observed on BC cores. The work provides valuable contribution into understanding the chemical behaviors and therefore the impacts on air quality and climate of OA. It can be accepted for publication in ACP, this reviewer has however a few minor comments as listed below:**

**(1) There are a few typos, grammar or format errors in the manuscript that should be corrected, for example, in Line 38, Line 67, Line 79, Line 235-236, Line299, etc.**

Thanks for the comment. The typos, grammar, and format errors in the manuscript mentioned above were corrected, and a thorough check is also conducted to correct other errors

**(2) Line 114: it is not clear what is the "BC-free species" referring to.**

Now we changed it to non-BC containing particles.

**(3) Line 114-116: Explain a bit more why HR-AMS can measure Type I and II, and SP-AMS for Type II and III.**

We added a sentence in Lines 117-121: "*NR-PM$_1$ can be quickly vaporized by the 600 ºC tungsten vaporizer of HR-AMS and be detected. The SP-AMS used here was equipped only with the Nd-YAG intra-cavity infrared laser (1064 nm) (tungsten vaporizer was physically removed), it can selectively detect BC-containing particles only, which include Type II and Type III species.*

**(4) Line 150-155: Did you perform corrections, for example, on elemental ratios, between the two AMS as you state there could be some mass spectral differences due to measurement schemes?**

Yes, scaling factors of 1.10 for H:C and 0.86 for O:C by Canagaratna et al. 2015 were applied.

**(5) Line 248: I suggest to delete this sentence.**

As suggested, this sentence is removed.

**(6) Line 280-282: As you state that HOA quantification might be influenced by the changes of collection efficiency. Can you explain a bit more about the possible influences of the collection efficiency on other OA factors?**

The possible influences on CE has been described in Lines 151-155:"*It should be noted that the BC quantification will not be affected by particle bouncing without the tungsten vaporizer, which could affect the CE in the standard HR-AMS measurements (Canagaratna et al., 2007). However, the CE will be governed by the overlap of the particle beam and laser beam (Lee et al., 2017;Massoli et al., 2015;Willis et al., 2014).*"

**(7) Line 320: Is it possible that the A-BBOA fraction (for example, <5%) in total NR-PM$_1$ is too low to be resolved by the PMF?**

Yes, this is possible. The A-BBOA might be included in the NR-PM$_1$ MO-OOA as described in Lines 321-323.

**(8) Figure 6: It is better to put the legends adjacent to the HR-AMS and SP-AMS plots directly in d, to make it clear.**

Done

**Reviewer #2**
Wang et al. compare the OA properties by parallel measurements using SP-AMS and HR-AMS respectively, in summer Beijing. The AMS technique is suitable for online quantification of OA and in particular SP-AMS can provide a unique piece of OA that coated on rBC cores. The findings are therefore unique and valuable to understand the OA composition and chemistry in megacities like Beijing. The overall interpretation of the data is reasonable and the paper is well written, I suggest its acceptance in ACP after the following minor issues are well addressed.

**(1) During the APHH campaign, another type of mass spectrometer (single particle mass spectrometry) was used to elucidate the OA properties too. Some studies should be included here to facilitate the interpretation.**

Thanks for the suggestion, we have added two references by using single particle MS techniques in the revised manuscript.

Chen, Y.; Cai, J.; Wang, Z.; Peng, C.; Yao, X.; Tian, M.; Han, Y.; Shi, G.; Shi, Z.; Liu, Y.; Yang, X.; Zheng, M.; Zhu, T.; He, K.; Zhang, Q.; Yang, F. Simultaneous measurements of urban and rural particles in Beijing – Part 1: Chemical composition and mixing state. Atmos. Chem. Phys., 20, 9231-9247, 10.5194/acp-20-9231-2020, 2020a.
Chen, Y.; Shi, G.; Cai, J.; Shi, Z.; Wang, Z.; Yao, X.; Tian, M.; Peng, C.; Han, Y.; Zhu, T.; Liu, Y.; Yang, X.; Zheng, M.; Yang, F.; Zhang, Q.; He, K. Simultaneous measurements of urban and rural particles in Beijing – Part 2: Case studies of haze events and regional transport. Atmos. Chem. Phys., 20, 9249-9263, 10.5194/acp-20-9249-2020, 2020b.

**(2) Some typos or citation formats do not follow the ACP style, please check and revise.**

We have carefully checked the manuscript.

**(3) In the instrumentation section, some necessary technical details are missing. For example, what is the m/z range of the OA mass spectra for HR-AMS and SP-AMS? Time resolution? Operation modes (V or W?) Is the tungsten vaporizer physically removed or turned off in SP-AMS?**

Thanks for the comment. The m/z range of the OA mass spectra for HR-AMS and SP-AMS reported in this study is across m/z 12-120 as described in Lines 161. The operation mode is described in Lines 153-154 as well. However, we added a statement for technical details in Lines 123-125: "*Briefly, the two AMS were operated under the mass quantify favorable mode (V-mode) with a time resolution of five minutes.*" And a statement in 117-121: "*NR-PM1 can be flash vaporization via the 600 ºC tungsten*

*vaporizer of HR-AMS and thus to be detected. For BC-containing particles, due to the SP-AMS equipped a Nd-YAG intra-cavity infrared laser (1064 nm) module, it can selectively detect BC-containing particles (Type II and Type III) with the tungsten vaporizer be moved."*

**(4) Xie et al (Atmos Environ 2019; 213:499-504) shows different PMF results from this study, is it because the datasets used for PMF analysis are different?**

There are two reasons for the difference of PMF results between this study and the one reported by Xie et al. One reason is caused by the different PMF inputs. For example, in Xie's report, only carbon clusters and OA mass spectra were used in the PMF analysis, while ion fragments from inorganic species (e.g., $SO+$, $SO+$, $NO+$, $NO2+$, $Cl+$, $HCl+$, $NH+$, and $NH2+$) were also included in the PMF analysis. Another reason is because of the amount of dataset, in Xie's study, there are only 9 days (from June 4 to 13), while there are 26 day (from June 4 to 30).

**(5) Line 316-319: The ABBOA is not separated in the HR-AMS dataset, is it likely because that the ABBOA contains more refractory components?**

Although this can be investigated further but we think it is unlikely. Most organics are non-refractory and there is no specific reason that BBOA contains more refracrtory components than other types of OA. In addition, according to previous studies of SP-AMS, the evaporation of non-BC species associated with BC core are under lower temperature. The reason why the A-BBOA was not separated in HR-AMS measure OA might be caused by its low mass fraction in the total OA (but not that low in BC-PM$_1$ OA), for example, less than 5% of total OA.

**(6) References: Line 504-516, the references are the same, but repeated twice.**

Corrected.

**(7) Figure 5: There are only four ion families here. How about the nitrogen-containing organic ions, although they may have little influences?**

The nitrogen-containing organic ion fragments were also involved in the PMF analysis as shown in the Figure 2, however, those signals are relatively very low, and for better comparison, nitrogen-containing organic ion fragments were not shown here.